# An Overview of SARS-CoV-2 Etiopathogenesis and Recent Developments in COVID-19 Vaccines

**DOI:** 10.3390/biom13111565

**Published:** 2023-10-24

**Authors:** Dona Susan Mathew, Tirtha Pandya, Het Pandya, Yuzen Vaghela, Selvakumar Subbian

**Affiliations:** 1Department of Microbiology, Amrita Institute of Medical Science and Research Centre, Amrita Vishwa Vidyapeetham, Kochi 608204, India; donsmathew30@gmail.com; 2Public Health Research Institute (PHRI) Center, New Jersey Medical School, Rutgers University, Newark, NJ 07103, USA; tsp77@scarletmail.rutgers.edu (T.P.); hsp1297@gmail.com (H.P.); ybv3@scarletmail.rutgers.edu (Y.V.)

**Keywords:** epidemiology, pathogenesis, viral vector, nucleic acid vaccine, protein subunit, vaccine trials

## Abstract

The Coronavirus disease-2019 (COVID-19), caused by the severe acute respiratory syndrome coronavirus-2 (SARS-CoV-2), has significantly impacted the health and socioeconomic status of humans worldwide. Pulmonary infection of SARS-CoV-2 results in exorbitant viral replication and associated onset of inflammatory cytokine storm and disease pathology in various internal organs. However, the etiopathogenesis of SARS-CoV-2 infection is not fully understood. Currently, there are no targeted therapies available to cure COVID-19, and most patients are treated empirically with anti-inflammatory and/or anti-viral drugs, based on the disease symptoms. Although several types of vaccines are currently implemented to control COVID-19 and prevent viral dissemination, the emergence of new variants of SARS-CoV-2 that can evade the vaccine-induced protective immunity poses challenges to current vaccination strategies and highlights the necessity to develop better and improved vaccines. In this review, we summarize the etiopathogenesis of SARS-CoV-2 and elaborately discuss various types of vaccines and vaccination strategies, focusing on those vaccines that are currently in use worldwide to combat COVID-19 or in various stages of clinical development to use in humans.

## 1. Introduction

Currently, humanity is passing through an outbreak of coronavirus disease-2019 (COVID-19), caused by a novel severe acute respiratory syndrome coronavirus-2 (SARS-CoV-2). The World Health Organization (WHO) declared COVID-19 as a global pandemic on 11 March 2020. This disease outbreak originated from a local seafood market with the first patient reported in the Wuhan province of China on 12 December 2019. With its ability to infect through the respiratory tract and transmit through aerosol, combined with the lack of fully effective vaccines or therapeutic interventions, COVID-19 is causing significant morbidity and mortality in humans worldwide. Although COVID-19 was initially thought to affect people above 60 years of age, with very less impact on children who are rarely affected or usually asymptomatic, more recent studies indicated that individuals with poor immune function and/or with comorbid conditions, such as diabetes, cardiovascular disease, chronic respiratory disease, cancer, renal, and hepatic dysfunction are at a higher risk for developing severe COVID-19 [1,2,3,4].

Coronaviruses are positive-stranded RNA viruses that have crown-like spikes on their surface (hence their name) and are a group of Nidovirus superfamily with three subtypes—α, β, γ and δ coronavirus. They cause diseases in animals (domestic and wild) as well as humans [5]. Respiratory viral infections in humans can vary from common colds to severe respiratory diseases. Most of the common human coronaviruses, including the 229E (alpha), NL63 (alpha). OC43 (beta), and HKU1 (beta), are associated with a wide range of upper respiratory tract infections [6]. There are so far three incidences of novel coronavirus-infected pneumonia (NCIP) reported by WHO—SARS-CoV (beta coronavirus that caused severe acute respiratory syndrome in 2003), MERS-CoV (beta coronavirus that caused Middle East Respiratory Syndrome in 2012), and COVID-19 or SARS-CoV-2 (beta coronavirus causing disease in 2019) [4].

In this review, we summarize the pathogenesis and treatment of COVID-19 and elaborately discuss various types of vaccines currently used worldwide to protect humans against COVID-19. 

## 2. SARS-CoV-2 Pathogenesis and Treatment of COVID-19

### 2.1. Clinical Features and Etiopathogenesis of SARS-CoV-2

SARS-CoV-2 primarily affects the respiratory system although other organs are also affected as disease progresses. Initially, respiratory tract infection-related symptoms, including fever, cough, sore throat, dyspnea, dizziness, and chest tightness were reported in the case series from Wuhan, China and this heterogeneity further expanded to acute respiratory distress syndrome (ARDS). Other symptoms observed were headache, generalized weakness, loss of smell and taste, as well as vomiting, diarrhea, body pain, and memory loss [1,7,8]. SARS-CoV-2 enters the human body by inhalation of virus particles mainly by respiratory aerosol droplets, which are produced when an infected person coughs, sneezes, talks, or even breathes heavily. These are also the primary modes of viral transmission. Other modes like direct contact with contaminated surfaces and fecal-oral routes have also been reported [9]. Upon entering the respiratory system, SARS-CoV-2 binds to the receptor, Angiotensin Converting Enzyme 2 (ACE2), present in epithelial and endothelial cells of the upper respiratory tract. ACE2 receptors are also present in other non-immune and immune cells present in various organs like the heart, brain, skeletal muscles, kidney, intestine, and endothelial cells. ACE2 is part of the Renin Angiotensin System (RAS), well known for circulatory homeostasis where it plays a major role in balancing the levels of Angiotensin II (AngII) and Angiotensin [1,2,3,4,5,6,7]. ACE2 is involved locally in multiple biological processes which include inflammation, angiogenesis, cell proliferation, memory, sodium and water reabsorption, thrombosis, and plaque rupture. Human ACE2 protein is a zinc metallopeptidase, which is a type I transmembrane glycoprotein and has a single extracellular catalytic domain that predominantly localizes at the plasma membrane. Primarily a membrane protein in its full length, it also exists in truncated soluble form with the role of the latter in COVID-19 unclear. The SARS-CoV-2 binds with membrane ACE2 receptors with strong affinity and the infected cells undergo conformational changes as soon as the virus fuses into cell membranes [10,11].

Apart from ACE2, studies have shown that ACE2 independent entry mechanisms do exist for SARS-CoV-2. The receptors involved include extracellular matrix metalloproteinase inducer CD147 (EMMPRIN), neuropilin-1 (NRP-1), dipeptidyl peptidase 4 (DPP4), AXL tyrosine-protein kinase receptor, and C-Type lectins, including CD209/L, CLEC4G, low-density lipoprotein receptor class A domain-containing protein 3 (LDLRAD3), and transmembrane protein 30A (TMEM30A). The impact of these alternative receptors on the ACE2-dependent SARS-CoV-2 entry is unclear [12,13,14]. Although SARS-CoV-2 primarily affects the lungs, studies have shown tissue damage and clinical symptoms in multiple organ systems like cardiovascular, gastrointestinal, neurological, musculo-skeletal, renal, ocular, endocrine, and cutaneous in humans [15,16,17]. There are four main proteins in SARS-CoV-2: the spike protein (S), membrane protein (M), Envelope protein (E), and Nucleocapsid protein (N). S protein plays a major role in the entry of viruses into humans [18]. The S protein consists of two subunits, S1 and S2. S1 helps in binding the receptor and S2 helps in the fusion of cell membranes [16]. As soon as the S1 subunit binds with the ACE2 receptor of the host cell, a transmembrane serine protease TMPRSS2 cleaves it from S2, which undergoes conformational change, and fuses with the host cell [19,20,21]. Alternately, in cells expressing low TMPRSS2, once engulfed into an endosome, SARS-CoV-2 escapes from the endosome by activation of their spike protein by cathepsins (non-specific low pH activated proteases) and further fusion of the host cell with viral membrane. In both mechanisms, the SARS-CoV-2 releases its RNA into the host cell cytoplasm, which translates to form two polyproteins that help in the formation of a replication translation complex in a double membrane vesicle. The viral RNA undergoes further replication to produce other accessory and structural proteins. All components then assemble and fuse with the cell membrane to release the virus. Complete virions capable of infecting other cells are assembled through the binding of N proteins to RNA molecules, which are then covered by E and M proteins [22]. Several SARS-CoV-2 proteins, including open reading frame 3b (ORF3b), ORF6, ORF7, ORF8, and the N protein, can trigger inflammation while inhibiting host antimicrobial responses, such as delaying the onset of type I (IFNα, IFNβ etc.) and type III (IFN λ, IFNε etc.) interferon responses against viral infection [23]. Thus, hyperinflammation, in combination with the lack of effective antiviral responses against SARS-CoV-2 early on during infection, promotes disease progression that makes the patients succumb rapidly to COVID-19 [16]. Adaptive immune responses, mediated by both T cell (not-associated with antibodies but T- cell driven cytokines/chemokines) and B cells (associated with antibodies, such as IgG, IgA, IgM etc.) resulting in cell-mediated immunity and humoral immunity, respectively, were observed in COVID-19 [16].

The pattern recognition receptors (PRR) of the host cells recognize the viral entry and exert two major responses—one mediated by interferons type I and type III and its associated genes along with cytokine secretion and the second one by leukocytes along with associated chemokine secretion. The SARS-CoV-2 non-structural proteins (NSPs 10, 13, 14, 15, 16) help to shield the viral RNA and mimic host cell RNA to evade the PRRs. They also help avoid immune sensing besides triggering cytokine secretion. Dysregulation in the function of myeloid cells (dendritic cells), as well as innate lymphoid cells (NK cells), is observed in acute respiratory distress syndrome. In severe cases, there are increased numbers of neutrophils and monocytes, specifically cluster of differentiation 14/16 positive (CD14^+^CD16^+^) inflammatory monocytes, Granulocyte macrophage colony-stimulating factor positive (GM-CSF^+^), and interleukin 6 positive (IL-6^+^) monocytes, along with a marked reduction of T-cells. In severe and critical COVID-19 cases, the CD8^+^ and CD4^+^ T cell counts were significantly reduced which could be attributed to T-cell exhaustion and results in disease progression. Exhausted T-cell responses have been reported in viral infections and malignancies and usually emerge during chronic infections. SARS-CoV-2 elicits a strong B cell response as evidenced by the robust and rapid expression of various antibodies, including IgM, IgG, and IgA, soon after infection that can be detected in the blood sample of infected individuals [1,24,25,26]. 

Initial clinical studies of COVID-19 patients indicated that the severe and critically ill had high levels of cytokines, predominantly IL-2, IL-6, IL-10, interferon-gamma (IFN-γ) inducible protein 10 (IP10), monocyte chemoattractant protein 1 (MCP-1), GM-CSF, and tumor necrosis factor-alpha (TNF-α), along of lymphopenia. Pulmonary immune cell infiltrations indicated severe inflammation, failure of effective cellular immune response and a cytokine storm in the patients with severe COVID-19 [25,27,28]. In addition, severe COVID-19 manifests as acute respiratory distress syndrome (ARDS) with elevated plasma proinflammatory cytokines, including interleukin 1β (IL-1β), IL-6, (TNF-α), C-X-C motif chemokine ligand 10 (CXCL10/IP10), macrophage inflammatory protein 1 alpha (MIP-1α), and chemokine (C-C motif) ligand 2 (CCL2), with low levels of interferon type I (IFN-I) in the early stage and elevated levels of IFN-I during the advanced stage of COVID-19. Studies from current COVID-19 pandemics have thrown light on the host and viral responses in these RNA viruses [1,3,8,16,24].

### 2.2. Variants of SARS-CoV-2

The SARS-CoV-2 viral genome can undergo mutations that can alter the virus’ pathogenic potential and better adapt to the infected host. Even single nucleotide polymorphisms (SNPs) can generate variants of SARS-CoV-2, which can drastically affect the virus entry into host cells, evade the immune system, and modify its transmissibility, severity of clinical manifestation, neutralization by antibodies, and host response to treatment and vaccines. SARS-CoV-2 variants of concern (VOC) strains included alpha, beta, gamma, delta, epsilon, and omicron as classified by the WHO, and contained SNPs in the receptor binding domains (RBDs) of S proteins. All except gamma VOC had increased transmissibility, with omicron having an increased risk of re-infection [29]. In contrast, SARS-CoV-2 variants of interest (VOI) strains show less severity on all aspects of viral adaptation. The WHO has described eight VOIs namely epsilon, zeta, eta, theta, iota, kappa, lambda, and mu [29,30]. The SNPs in these VOCs and VOIs have been described previously [31,32]. The WHO has recently added XBB.1.5 and 1.16 as VOIs and new sub-lineages of omicron lineage strains, BA.2.75, CH 1.1 XBB 1.91, XBB 2.3, and BA 2.86 as Variants Under Monitoring (VUM) [33].

### 2.3. Treatment of COVID-19 

Although multiple SARS-CoV-2 strains can cause COVID-19 with different levels of severity in humans worldwide, there are not many effective treatment options available. While Molnupiravir and Paxlovid are available as orally administrable drugs (Appendix A), Remdesivir is prescribed for treating hospitalized COVID-19 patients. Paxlovid is a combination of Nirmatrelvir and Ritonavir. While Nirmatrelvir is a protease inhibitor, Ritonavir, a strong cytochrome P450 (CYP) 3A4 inhibitor and pharmacokinetic boosting agent increases the concentration of Nirmatrelvir to target therapeutic range. However, with Paxlovid, dose adjustment is required for patients with mild to moderate renal or hepatic impairment. Both Remdesivir and Molnupiravir are nucleoside analog prodrugs that do not need stringent dose adjustment in patients with renal and hepatic impairment. Although studies have shown that the use of oral drugs greatly reduced hospitalization of patients and reduced death, the use of corticosteroids was recommended along with anti-viral drugs for patients with severe COVID-19 [34,35,36].

In addition to the chemical drugs, monoclonal neutralizing antibodies namely Bebtelovimab, Tixagevimab, Cilgavimab, Bamlanivimab, Casirivimab, Imdevimab, Etesevimab, and Sotrovimab are approved by the USA-FDA for use in the treatment of COVID-19. These monoclonal antibodies target the receptor binding domain (RBD) of the viral S protein to tackle the progression of SARS-CoV-2 infection into COVID-19 and neutralize SARS-CoV-2 entry by blocking its engagement with the host cell receptors, such as the ACE2 receptor (Appendix A). However, the antibody therapy requires intravenous administration under medical supervision and is specifically used for high-risk groups. Though they are administered at the recommended dose, the patients may run the risk of immune-mediated reactions due to reactions to antibodies or off-target effects [36,37,38].

## 3. COVID-19 Vaccine Development

With the availability and suitability of fewer oral antiviral drugs and monoclonal antibodies to treat COVID-19, the world has quickly realized the urgent need for safe and effective vaccines. Vaccine research is being conducted to develop interventions that can prevent infection and/or reduce disease transmission. Vaccinations are the top priority for managing and controlling COVID-19, and many technological platforms are being investigated to generate effective vaccinations. Soon after the publication of the SARS-CoV-2 sequence on 11 January 2020, there was an international response to preparing the outbreak and this triggered the development of vaccines within a short period. Since SARS-CoV-2 is 79% genetically similar to SARS-CoV and MERS virus, and uses the same receptor, ACE2, to infect the host cells, efforts for developing a vaccine were initiated much earlier before declaring COVID-19 as a pandemic [39]. Many multinational pharmaceutical companies collaborated with governments worldwide and invested billions of dollars in developing vaccines. A successful COVID-19 vaccine requires a cautious validation of efficacy and adverse reactivity as the target vaccine population includes high-risk individuals over the age of 60 with comorbidities [39].

### 3.1. Characteristics of a COVID-19 Vaccine

The WHO released a target product profile (TPP) for COVID-19 vaccines in 2022 that provided guidelines for the development of COVID-19 vaccines (Table 1). The TPP includes preferred and minimally acceptable profiles for two human vaccines—the first one was intended for use in the long-term protection of persons at high ongoing risk of COVID-19 such as healthcare workers. And, the second was intended for reactive use in outbreak settings with rapid onset of immunity. Both were targeted for use against severe or critical COVID-19 cases, with the former targeting a broader target population range and the latter specifically for use in emergencies [40].

Table 1 shows the WHO’s target product profiles for vaccines in long-term protection compared to vaccines in reactive use [40].

### 3.2. Types of COVID-19 Vaccine

Development of vaccines involves multiple approaches either using the whole virus or parts of the virus, including the Deoxyribonucleic acid (DNA) or messenger ribonucleic acid (mRNA) or protein subunits (antigens) in every vaccine type, or the use of viral vectors for delivery. Every vaccine type aims at targeting antigen presentation through different mechanisms, with the ultimate goal of triggering immunity in the host as summarized in Figure 1.

#### 3.2.1. Whole Virus Vaccines

In this type, whole virus is inactivated or killed using chemical or physical agents such as β-propiolactone, formaldehyde, heat, or radiations. These vaccines can trigger primarily an antibody-mediated humoral immunity upon entry into the host and the immunity increases with repeated doses of vaccine administered. Importantly, the virus or the genetic material does not replicate inside the vaccinated host. Inactivated vaccines can be improved by using adjuvants and are favourable for long term protection against viral infection. In contrast, live attenuated vaccines use SARS-CoV-2 mutant subtypes with low pathogenicity. This weakened virus particles can replicate within the host cells and elicit both humoral and cell-mediated immune responses [41,42]. Currently, there are twenty-two inactivated viral vaccines in clinical trials with thirteen of them in phases 3 and 4. Among the live attenuated vaccines, two are in clinical trials and one in phase 3 (Table 2 and Appendix A).

There are eleven vaccines approved for use across different countries, and three of them—Covaxin, Covillo and Coronavac are also approved for emergency use. Covaxin is Bharat Biotech’s Whole Virion Inactivated Corona Virus Antigen BBV152, which is adjuvanted with aluminum hydroxide gel and/or Imidazo quinolin gallamide (IMDG), a Toll-like receptor (TLR) 7/8 agonist. Studies have shown that the vaccine is safe with milder and lesser adverse events and 93.4% efficacy against severe COVID-19 in phase 3 trials. The vaccine has been shown to induce neutralizing antibodies and T-cell response to protect up to 6 months against alpha, beta, and delta variants of SARS-CoV-2 but not against the omicron variants [43,44,45,46]. While CoronaVac was prepared using SARS-CoV-2 (CZ02 strain), Covilo was prepared using the SARS-CoV-2 19nCoV-CDC-Tan-HB02 strain with aluminum hydroxide as an adjuvant.

This vaccine was proven to be safe and efficacious against severe cases of COVID-19, by inducing neutralizing antibodies against alpha and beta but at lower levels against delta variants of SARS-CoV-2 [47,48,49]. Additionally, some of these vaccines were shown to offer protection against the omicron variant in affected COVID-19 cases including children [50]. Although inactivated and live attenuated virus has all the components of the original SARS-CoV-2 to generate strong antibody and T cell responses and produce long-lasting immunity, it requires very careful preparation and storage methods and care with use in immunocompromised individuals.

#### 3.2.2. Component Viral Vaccines

Component viral vaccines include protein subunit vaccines, nucleic acid vaccines, viral-like particles, and viral vector vaccines [51].

##### Protein Subunit Vaccines

Protein subunit vaccines are made up of one or more fragments or antigens of the virus that can trigger an immune response and offer protection against SARS-CoV-2 infection. Owing to its inability to display the full antigenic complexity of the virus, component vaccines are generally considered to be safe. However, with limitations to only selective recombinant antigens of interest, immunogenicity conferred by the component vaccines would be poor and needs adjuvant to improve protective efficacy against SARS-CoV-2 infection. There are 59 protein subunit vaccines in various phases of clinical trials with 14 of them in phases 3 and 4 as of 2022 (Table 2 and Appendix A). Nineteen of these protein subunit vaccines are approved for use across different countries, and two of them—Nuvaxovid (Novomax) and COVOVAX (Serum Institute of India) are recommended for emergency use (Appendix A). Novomax contains a recombinant spike glycoprotein of SARS-CoV-2 as a nanoparticle, combined with saponin-based Matrix M1 as an adjuvant, and has been used in 40 countries. Novomax is safe with 89.7% efficacy in adults and neutralizing antibodies reported to offer protection against infection by the alpha, beta, delta, and omicron variants of SARS-CoV-2 [52,53,54,55]. Another widely used bivalent protein subunit vaccine was developed by Sanofi/Glaxo Smith Kline using the S protein of the ancestral D614 (Wuhan strain) and the beta (B.1.351) variant of SARS-CoV-2, combined with the GSK AS03 adjuvant system (CoV2 preS dTM-AS03). Immune protection of this vaccine was through the production of virus-neutralizing antibodies, which have shown safety and offer a protective efficacy of 75.1% against delta and omicron variants [56,57].

##### DNA Vaccines

DNA vaccines are plasmid-based delivery systems that carry a plasmid DNA encoding the viral protein, such as the S protein of SARS-CoV-2. Once injected into the host, the viral proteins are expressed and undergo appropriate species-specific posttranslational modifications to trigger an immune response against the infecting virus. DNA vaccines usually trigger both humoral and cell-mediated immune responses if delivered using adjuvants or nanoparticle-based delivery systems, but still have low immunogenicity warranting repeated doses. However, to their advantage, DNA vaccines are safe, robust in production, and relatively stable for storage and transportation [41,56,58]. Currently, there are two DNA vaccines in phase 3 trials and 17 in total under various stages of clinical trials (Table 2 and Appendix A). There is only one DNA vaccine, ZyCoV-D, developed by Zydus Cadila, India, approved for use in long-term protection but not for emergency use. ZyCoV-D contains the DNA for the S protein of the Wuhan strain of SARS-CoV-2 Hu-1, along with an IgE signal peptide carried in a pVax1 delivery plasmid. This vaccine was considered safe with an efficacy of 66.6% in symptomatic patients, including children and adults. The vaccine was reported to be efficient against several variants of SARS-CoV-2, up to the delta variant [41,59,60]. The DNA vaccines were delivered intradermally; currently, the administration of these vaccines is moved from a needle injection system to a needle-free injection system with improved immunogenicity and protective efficacy mediated by both the humoral and cell-mediated immunity [61,62].

DNA vaccines have good stability and are considered safer to handle and are relatively inexpensive to generate. However, antibody and T cell titers are low with DNA vaccines, compared to the protein vaccines, due to the integration of the former vaccines with the host genome. There are also safety concerns that DNA vaccines may persist in the body for a long period and result in host cell mutations, leading to the development of tumors or malignancies.

##### mRNA Vaccines

The logic behind the mRNA vaccines is to deliver specific SARS-CoV-2 mRNA, such as the one that codes for S protein, to be translated in the host cell. The viral protein thus formed will be further processed by the host proteosome mechinery for presentation to the immune system to elicit humoral and cellular-immunity against subsequent infection [61,62]. Current mRNA vaccines for COVID-19 can be classified into three types namely non-replicating mRNA, self-amplifying mRNA (saRNA), and circular RNA (circRNA). While the non-replicating mRNA uses the coding region of the target protein alone, saRNA uses the genetic portion of antigen along with another RNA viral genome for delivery and increased antigenic response and the circRNA uses the non-coding RNA of eukaryotic cells to create a covalently closed loop structure along with viral antigen region into stable circular RNA. However, most of the currently approved mRNA vaccines are of the non-replicating mRNA type. The SARS-CoV-2 mRNA is negatively charged and unstable thus requiring a delivery vehicle, such as a lipid nanoparticle, lipopolyplexes, polymeric nanoparticles, and cationic polypeptides. These mRNA vaccines can be administered intradermally (ID), intramuscularly (IM), and/or subcutaneously [61,63]. There are forty-three mRNA vaccines in clinical trials with five in phases 3–4 clinical trials (Table 2 and Appendix A). Currently, nine mRNA vaccines are approved for use in long-term protection with two of them—Spikevax and Comirnaty recommended for emergency use. The Spikevax (mRNA-1273) was developed through a collaborative effort between Moderna and the Vaccine Research Center at the US-National Institute of Allergy and Infectious Diseases (NIAID). In this vaccine, the SARS-CoV-2 S protein is locked in a pre-fusion conformation offering improved stability and immunogenicity and combined with lipid nanoparticles for delivery. Spikevax was safe with an efficacy of 94.1% among severe COVID-19 cases [64,65]. Spikevax induced primarily humoral immune response with poor T-cell involvement, although it provided efficient protection against even the omicron variant [65,66]. Comirnaty is another widely used mRNA-based vaccine against COVID-19, developed by BioNTech, Germany in collaboration with Pfizer, USA. This mRNA vaccine carries the S protein mRNA in a lipid nanoparticle delivery vehicle. The vaccine is proven safe with an efficacy of 95% and efficient in protecting against infection by the omicron variants [67,68,69]. The vaccine induces robust cell-mediated immunity and additional humoral response for long-term protection [66]. While mRNA vaccines are faster to manufacture, they have a limited shelf life and may require multiple/booster doses to sustain the protection.

##### Virus-Like Particles (VLPs)

Viral-like particles (VLPs) contain virus multimeric particles resembling the native virus but are non-infectious and do not contain any genetic material of the infectious virus. The multimeric VLPs are used to generate humoral, and cell-mediated immunity and use adjuvants to improve their immunogenicity. There are two types of VLPs—enveloped (eVLPs) and non-enveloped (neVLPs), the former uses the host membrane portions, including the glycoprotein of the virus, to trigger an immune response [70]. There are seven VLPs in clinical trials with three of them in Phase 3, with one VLP approved for use in long-term protection against SARS-CoV-2 infection (Table 2 and Appendix A). The Covifenx is a VLP produced in a plant-based platform, where the expression of the S protein generated VLPs that were subsequently adjuvanted with AS03. This vaccine induced both neutralizing antibodies and cell-mediated immunity with the generation of IFN-gamma and IL-4 [71]. The vaccine was safe with an efficacy of 78.8% against moderate to severe COVID-19 [72]. In general, VLPs are faster to manufacture and relatively stable for a long time with minimal risk of use in immunocompromised individuals.

##### Viral Vectors

Usually, the S or any target protein of SARS-CoV-2 is inserted into the carrier viruses to generate a viral vector. Viral vectors deliver the target antigen to the host cell where a long-term humoral and cell-mediated immunity against SARS-CoV-2 infection is elicited. So, this delivery system is more efficient than a non-viral protein or nucleic acid-based vaccine system. Viral vectors can be replicative or non-replicative where the non-replicative viral vectors (NRVVs) have better safety profiles and replicative viral vectors (RVVs) have greater immunogenicity. Several types of human and animal viruses, including Adenovirus of human and animal origin, the measles virus, vaccinia Ankara, the vesicular stomatitis virus, and the cytomegalovirus have been modified for use as vector-based vaccines against COVID-19. Though the human adenoviruses Ad5 was the first choice for developing a viral vector-based vaccine for COVID-19, this vector was reported to elicit a higher seroprevalence rate in the human population; thus, the use of alternates like Ad26 and Ad35 was encouraged. Besides, primate adenoviruses, such as the chimpanzee adenoviruses (ChAdOx1, cAD3, cAD68, cAD36) and gorilla adenovirus serotype 32 (GRAd32), as well as the modified Ankara virus, paramyxoviruses, and vesicular viruses were also used as viral vectors [42,73,74]. There are twenty-six non-replicating and six replicating viral vectors currently in clinical trials with seven of the former in phases 3/4 and one of the latter in phase 3 trials (Table 2 and Appendix A). There are nine NRVVs approved for use in long-term protection against SARS-CoV-2 infection, of which four were also recommended for emergency use. The iNCOVACC (BBV154) was a chimpanzee adenoviral-vectored (cAd36) vaccine developed by Bharat Biotech International Limited (BBIL), India, encoding prefusion-stabilized S protein with two proline substitutions in the S2 subunit. This vaccine is administered intranasally and proven safe and efficacious and elicits robust humoral and cell-mediated immunity with protection against even the omicron variants [75]. The Convidecia developed by CanSino Biologics and the Beijing Institute of Biotechnology, is a single-dose adenovirus Ad5 vectored vaccine expressing the SARS-CoV-2 S protein. This vaccine elicited both humoral and cell-mediated immunity and had 96.0% efficacy after 14 days of vaccination but no data on its protective effect on SARS-CoV-2 variants [76]. Convidecia Air is similar to the Ad5-SARS-CoV-2 vaccine but given by inhalation and developed and approved in China. This vaccine elicited an increased level of bronchial IgA levels [77]. The Sputnik V (Gam-COVID-Vac) was developed at Gamaleya National Research Centre for Epidemiology and Microbiology (Moscow, Russia). It uses a heterologous recombinant adenovirus approach with Ad26 and Ad5 as vectors for the expression of the S protein. This vaccine was safe with 91.6% efficacy and generated a good humoral and cell-mediated immunity against SARS-CoV-2, including the omicron variants [78,79,80,81]. Sputnik light is another COVID-19 vaccine by Gamaleya where the S protein is expressed by the Ad5 vector. This vaccine has an efficacy of 79.4% and has proven effective against all variants including delta and omicron [81,82,83]. The Ad26.COV2.S or Jcovden was developed by Janssen (Beerse, Belgium), the pharmaceutical wing of Johnson & Johnson (JNJ). This vaccine is an Ad26-based vector containing prefusion-stabilized S protein (furin cleavage site mutation and proline substitution) for better immunogenicity and was safe with efficacy of 73.1% and 81.7% against severe and critical COVID-19 cases at 14- and 28-days post-vaccination [84,85]. The Vaxzevria (AZD-1222) or ChAdOx1 vaccine is composed of a ChAdOx1 chimpanzee adenovirus vector developed in the United Kingdom by Oxford University and AstraZeneca. This vaccine is safe with an efficacy of 66.7% at 14 days post-vaccination and elicited both humoral and cell-mediated immunity with induction of IFN-gamma and TNF-alpha along with CD8+ T cells among different SARS-CoV-2 variants, including omicron [86,87,88,89,90]. The same vaccine has also been manufactured by the Serum Institute of India as Covaxin and provided similar immunogenicity and protective efficacy as ChAdOx-1.

The production of viral vectors is challenging since they require special facilities to produce and maintain their purity; there is a high risk of mutation of the virus. Besides, these vaccines are not suitable for use in immunocompromised/immunosuppressed individuals as the immune system is unable to contain slow replication of the viral vectors and needs to be carefully administered. The pros and cons of various types of COVID-19 vaccines are presented in Table 3.

### 3.3. Mix and Match Concept

Although homologous (same-type) vaccines are applied as primary and booster doses, there are studies on the use of heterologous (different-type) vaccine regimens to prevent COVID-19. This concept also includes the interchanging use of mRNA vaccines with adenoviral-vector vaccines as prime and boosters or vice versa. This approach has been rationalized to improve immunogenicity and safety as well as to mitigate intermittent supply shortages of COVID-19 vaccines. Heterologous priming with the Vaxzevria followed by boosting with Spikevax was tried and tested in many studies [91]. In a phase 2 CombiVacS trial in Germany, adults aged 18–60 years were given ChAdOx1-S as prime and BNT162b2 as booster vaccines at 8–12 weeks apart. This study showed that heterologous prime-boosting vaccination induced a robust immune response, with an “acceptable and manageable reactogenicity profile” [92]. Interestingly, the heterologous CombiVacS vaccination with 10–12 weeks intervals were shown to improve the immunogenicity compared to homologous vaccination with individual vaccine components (i.e, ChAdOx1 and BNT162b2) [93]. In the Com-COV inferiority trial conducted in the UK, participants of 50 years or older age were vaccinated with one of the following four combinations as prime/booster regimen: ChAdOx1/ChAdOx1; ChAdOx1/BNT162b2; BNT162b2/BNT162b2 or BNT162b2/ChAdOx1 at four week interval between vaccination [94]. Results of this trial showed that the heterologous BNT162b2/ChAdOx1 vaccination group had a higher anti-S protein antibody response, while ChAdOx1/BNT162b2 showed a higher immunogenicity, compared to the standard homologous ChAdOx1/ChAdOx1 vaccination. In Argentina, the limited availability of the Sputnik V vaccine prompted a heterologous vaccination approach with Sputnik V as prime, followed by Moderna as a booster vaccine in adults. In this case, the adverse events, including local and systemic symptoms were reported to be higher among heterologous SpV/Mod than for homologous SpV/SpV vaccination. However, the study showed that the heterologous immunization with SpV/Mod is not only more immunogenic and well-tolerated but also induces a stronger humoral response compared to the homologous SpV/SpV vaccination [95]. In a non-inferiority phase 4 trial in India, heterologous prime/boost was attempted using Covishield and Covaxin as prime/boost vaccines, respectively. In the study, adults aged above 18 years were vaccinated with Covishield and Covaxin at four weeks intervals and the results showed that the heterologous approach showed acceptable reactogenicity in the first seven days after booster vaccination. Interestingly, Covishield as a booster after Covaxin prime produced robust immunogenicity, while Covaxin as a booster after Covishield prime induced a weaker immunogenicity yet substantiating the levels of acceptance as per existing WHO recommendations, thus encouraging Covishield to be used as a booster, irrespective of the nature of the prime vaccine [96].

Taken together, these data highlight the flexibility of applying the heterologous vaccination strategy without compromising the safety, immunogenicity, and protective efficacy of individual vaccines. A summary of various COVID-19 vaccine types in clinical development is presented in Figure 2.

### 3.4. Routes of Vaccination

The most common route of administration of COVID-19 vaccines is intramuscular injection, and most of the approved vaccines (47 out of 50) currently use this route. However, other routes, including non-invasive inhalation exist as options for COVID-19 vaccination. ZyCoV-D, a DNA plasmid-based vaccine developed in India, is the only vaccine approved for the intradermal route of vaccination using a jet injector [97]. Another route of vaccination is through inhalation; Convidecia Air, which is derived from the adenovirus vector-based intramuscular version of Convidecia, produced by CanSinoBio is approved in China; this route was reported to be safe, well-tolerated and elicited a strong mucosal, humoral, and cellular immunity [98,99]. In Phase 3 clinical trial, the intranasal vaccine for COVID-19 (iNCOVACC), which is an adenoviral vectored BBV154 vaccine, developed in India was shown to stimulate a broader immunity eliciting strong mucosal, humoral and cell-mediated host responses [100]. These findings show that various types of COVID-19 vaccines, including NRVV, DNA vaccines, whole viral vaccines, and protein subunit vaccines may be administered through routes alternative to conventional intramuscular injection, such as intradermal, intranasally, or inhalation, without compromising the beneficial immunogenicity and protective efficacy against SARS-CoV-2 infection. The COVID-19 vaccines administered through different routes are summarized in Figure 3.

## 4. Summary and Conclusions

Since the onset of the pandemic in early 2020, there have been about 771 million confirmed cases of COVID-19 globally, including about seven million deaths reported to the WHO [101]. As of 22 September 2023, a total of about 13.5 billion vaccine doses have been administered worldwide [101]. Various types of vaccines, based on whole-virus, viral vectors, nucleic acids and proteins have been currently in use worldwide and additional vaccines are under different stages of development. The central objective of vaccination is to enhance the host protective immunity against progression of initial SARS-CoV-2 infection into symptomatic clinical disease (Appendix A). Various types of vaccines differentially elicit the innate immunity and the adaptive immunity, including the humoral and cell-mediated immunity against SARS-CoV-2 infection. This reflects in the disparity in overall protection of vaccinated individuals. Further, the shorter durability of vaccine-induced immunity against SARS-CoV-2 necessitates the administration of additional booster doses of the vaccines within short span of time. This is a concern for vaccine compliance and implementation across various populations worldwide. In addition, new variants of SARS-CoV-2 with variable degrees of virulence and pathogenicity are emerging, which pose significant challenges to the vaccine-induced immunity to protect against new infections. To tackle this issue, administration of newer vaccine derivates as well as additional booster doses of vaccines have been mandated in many countries, while annual and/or seasonal vaccination approaches have also been considered. Additional research and preclinical testing are necessary to develop more efficacious vaccines that can protect against infection by a range of SARS-CoV-2 variants. New and novel vaccines are also needed to effectively vaccinate individuals with compromised immune systems and children. Alternate strategies such as heterologous vaccine administration as well as different routes of vaccine administration should also be improvised and evaluated for a wider vaccine application. Clearly, the management of COVID-19 through effective vaccination is not fully streamlined; refinement and optimization of strategies at the level of scientific, administrative, and political levels are required to make progress in this arena. With a comprehensive vaccination and disease management plan, the threat of COVID-19 can be better mitigated worldwide.

## Figures and Tables

**Figure 1 biomolecules-13-01565-f001:**
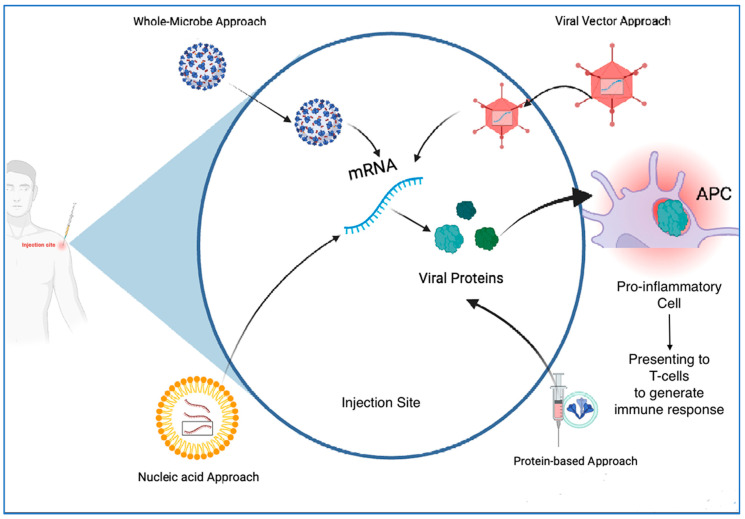
Summary of different vaccine types for COVID-19 and their role in immunity: Summary of different vaccine types including whole virus, nucleic acid, protein subunit, virus-like particles (VLP), and viral vectors with their role in mounting an immune response in the vaccinated host to protect against the development of COVID-19. (Created with Biorender.com, accessed on 22 September 2023). APC—antigen-presenting cells.

**Figure 2 biomolecules-13-01565-f002:**
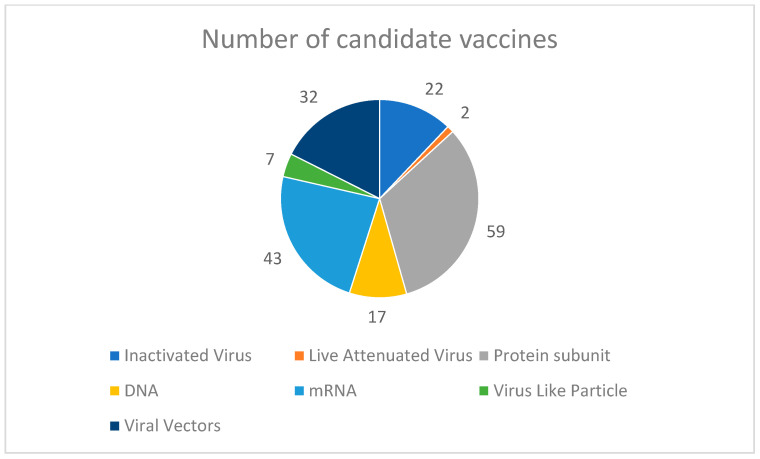
Summary of the spectrum of COVID-19 vaccine type currently in clinical development. This pie-chart indicates the spectrum of COVID-19 vaccines currently in the pipeline of clinical development [51]. Data labels indicate the actual number of vaccines under development in the corresponding type.

**Figure 3 biomolecules-13-01565-f003:**
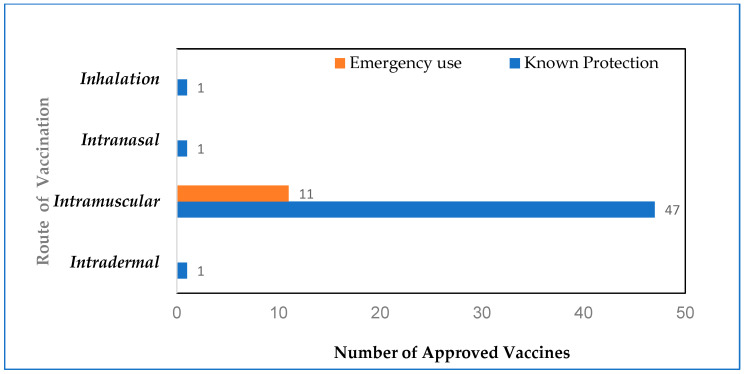
Number of COVID-19 vaccines administered by different routes: The chart indicates the distribution of different routes of administration for the COVID-19 vaccines currently approved by WHO and includes inhalation, intranasal, intramuscular, and intradermal routes with data labels representing the actual number of vaccines in each route.

**Table 1 biomolecules-13-01565-t001:** The WHO target product profile guideline for COVID-19 vaccines.

Characteristics	Vaccine for Long-Term Protection	Vaccine for Reactive Use
Indications	Use in long-term protection of persons at high ongoing risk of COVID-19; Potential for administration with other vaccines.	Reactive use in outbreak settings with rapid onset of immunity; Stand lone administration is acceptable.
Target population	Adult and children	Adult
Contraindications	Minor	Contraindications accepted in some conditions
Safety	Substantial evidence required	Acceptable if it outweighs potential risk
Dose regimen	Single or double dose along with mix and match options.	Double dose is preferred.
Durability	At least 1 year before use of another booster dose	Until protection from severe disease.
Route of administration	Preferably nasal or oral with no use of needle/syringe	Any suitable mode
Storage	Higher thermostability is preferred with shelf life of preferably 2 months at 2–8 °C.	Stability in deep freezer with shelf life of at least a month at 2–8 °C.
Number of vaccines approved by WHO (as of August 2023)	50 (47 administered intramuscularly)	11 (all administered intramuscularly)

**Table 2 biomolecules-13-01565-t002:** COVID-19 vaccines currently in clinical phase 3/4 trials worldwide.

Type of Vaccine	Clinical Trials	Phase 4	Phase 3
Inactivated	22	3	10
Live attenuated	2	NIL	1
Protein subunit	59	1	23
DNA Vaccines	17	NIL	2
mRNA vaccines	43	3	7
Virus-like particle	7	NIL	3
Non-replicating viral vector	26	4	3
Replicating viral vector	6	NIL	1

**Table 3 biomolecules-13-01565-t003:** Summary of the pros and cons of various types of COVID-19 vaccines.

S. No.	Type of Vaccine	Pros	Cons
1	Inactivated virus	Safe and easy to prepare Native antigen expression Stronger immune response Storage is easier	Requires adjuvants Administration in immunocompromised patients
2	Live attenuated virus	Stronger immune response Native antigen expression	Safety especially administration in immunocompromised patients Storage issues Requires adjuvants
3	Protein subunit vaccine	Safe and well-tolerated Good stability	Moderate immunogenicity Requires adjuvants
4	DNA vaccine	Good stability Safe to handle Relatively inexpensive to generate	Moderate immunogenicity Requires adjuvants or nanoparticles Need repeated doses Risk of integration into host genome
5	mRNA vaccine	More research and development data Simple production process No risk of integration into the host DNA Strong immunogenicity	mRNA is unstable and easily degraded Storage issues Requires adjuvants or nanoparticles Need repeated doses.
6	Virus-like particles (VLPs)	Easier to manufacture Good stability No risk for use with immunocompromised individuals	Moderate immunogenicity Requires adjuvants or nanoparticles Need repeated doses
7	Viral vectors	Rapid research and development Strong immunogenicity Long-term protection	Risk of pre-existing immunity against the viral vector used Expensive to produce

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
