# Peer review of "An Overview of SARS-CoV-2 Etiopathogenesis and Recent Developments in COVID-19 Vaccines"

_biomolecules, 2023, doi:10.3390/biom13111565_

Round 1

Reviewer 1 Report

The authors have done a good job in compiling the details on SARS cov2 infection and cure, however some comments (attached) needs to be addressed

Author Response

Comment: I appreciate reading this work, which could be interesting. However, the mechanisms mentioned is very general.  What other organs? What does the infection depend on? of the receptors that are expressed? Are only ACE2 receptors involved in SAR-CoV-2 infection? (It will be possible to merge paragraphs, Line 60-74)

Response: We thank the reviewer for the positive note in our review. With regard to other organs, we have added “Although SARS=CoV2 primarily affects the lungs, studies have shown tissue damage and clinical symptoms in multiple organ systems like cardiovascular, gastrointestinal, neurological, musculoskeletal, renal, ocular, endocrine and cutaneous in humans” to lines 90-92 with appropriate citation. We have added the details on alternate receptors as follows “Apart from ACE2, studies have shown that ACE2 independent entry mechanisms do exist for SARS-CoV2. The receptors involved include extracellular matrix metalloproteinase inducer CD147 (EMMPRIN), Neuropilin-1 (NRP-1), Dipeptidyl peptidase 4 (DPP4), AXL, tyrosine-protein kinase receptor, C-Type Lectins CD209/L; CLEC4G, Low-density lipoprotein receptor class A do-main-containing protein 3 (LDLRAD3), transmembrane protein 30A (TMEM30A). The impact of these alternative receptors on the ACE2-dependent SARS-CoV2 entry and levels of COVID-19 disease is unclear” in lines 84-89 with appropriate citation,
We have merged paragraphs as suggested (lines 60-74)

Comment:  The text contains repeated abbreviations. For example Line 76 and line 86.

Response: We have removed the repeated abbreviations as suggested in lines 93 and 104

Comment: What cytokines are involved? Line 88

Response: We have mentioned the cytokines involved as “Initial reports from COVID-19 patients indicated that the severe and critically ill had high levels of cytokines, predominantly IL-2, IL-6, IL-10, interferon-gamma (IFN-γ) inducible protein 10 (IP10), monocyte chemoattractant protein 1 (MCP-1), GM-CSF and tumor necrosis factor-alpha (TNF-α) along of lymphopenia. Pulmonary immune cell infiltrations indicated severe inflammation, failure of effective cellular immune response, and a cytokine storm in the patients with severe COVID-19 [22,24,25]. In addition, severe COVID-19 manifests as acute respiratory distress syndrome (ARDS) with elevated plasma proinflammatory cytokines, including interleukin 1β (IL-1β), IL-6, (TNF-α), C-X-C motif chemokine ligand 10 (CXCL10/IP10), macrophage inflammatory protein 1 alpha (MIP-1α), and chemokine (C-C motif) ligand 2 (CCL2), with low levels of interferon type I (IFN-I) in the early stage and elevated levels of IFN-I during the advanced stage of COVID-19” in lines 130-139.

Comment:  Abbreviations are also missing, for example GM-CSF (lines 99 or 106). Name abbreviations and their meaning first, and not just the abbreviation.For example, TNF-α from line 106 and tumor necrosis factor α (TNF-α) from line 110

Response: We apologize for this mistake. We have now added the abbreviations and their meanings as suggested the reviewer between lines 122 and 137 and 132-136.

Comment: You can make a table of Bebtelovimab, Tixagevimab, Cilgavimab, Bamlanivimab, Casirivimab, Imdevimab, Etesevimab, and Sotrovimab, showing their main characteristics/ Adverse effects/ differences/similarities that facilitate understanding of the mechanism of each vaccine on the body on the body in the treatment of covid-19.

Response:  We appreciate the suggestion by this reviewer. As suggested, we have included a new Supplementary Figure-2 to summarize various antibodies and their characteristics and adverse effects. The detailed explanation of the mode of action of each of these antibodies, their similarities and differences will be presented as a separate review in the future, since that is too much to add into this already comprehensive review.

Comment: Specify the nucleic acids or proteins used in the COVID-19 vaccine, as well as vehicles or excipients included. Line 204

Response:   We have now rephrased the sentence to include the details as “Development of vaccines involves multiple approaches either using the whole virus or parts of the virus, including the Deoxyribonucleic acid (DNA) or messenger ribonucleic acid (mRNA) or protein subunits (antigens) in every vaccine type, or the use of viral vectors for delivery” in lines 232-234

Comment:  Figure 1 should be more specific to the covid-19 vaccine, it is understood that it refers to any type of vaccine

Response:   We apologize for the lack of clarity in this sentence. As suggested by the reviewer, we changed the title as “Summary of different vaccine types for COVID-19 and their role in immunity”. The components of Figure-1 and the legend are focused towards COVID-19.

Comment: More organization is necessary in the text in general.

Response:   We thank the reviewer for this suggestion. We have revised the entire review as per the suggestions from all the reviewers.

Reviewer 2 Report

I appreciate reading this work, which could be interesting. However, the mechanisms mentioned is very general.

·      What other organs? What does the infection depend on? of the receptors that are expressed? Are only ACE2 receptors involved in SAR-CoV-2 infection? (It will be possible to merge paragraphs, Line 60-74)

·       The text contains repeated abbreviations. For example Line 76 and line 86.

·       What cytokines are involved? Line 88

·       Abbreviations are also missing, for example GM-CSF (lines 99 or 106)

·       Name abbreviations and their meaning first, and not just the abbreviation. For example, TNF-α from line 106 and tumor necrosis factor α (TNF-α) from line 110

·      You can make a table of Bebtelovimab, Tixagevimab, Cilgavimab, Bamlanivimab, Casirivimab, Imdevimab, Etesevimab, and Sotrovimab, showing their main characteristics/ Adverse effects/ differences/similarities that facilitate understanding of the mechanism of each vaccine on the body on the body in the treatment of covid-19.

·       Specify the nucleic acids or proteins used in the COVID-19 vaccine, as well as vehicles or excipients included. Line 204

·       Figure 1 should be more specific to the covid-19 vaccine, it is understood that it refers to any type of vaccine

·       More organization is necessary in the text in general.

Author Response

Comment: The authors summarize the cause and treatment for SARS cov2, identified to date. There are a few comments that need to be addressed. Line 70- The number “2” has to be added after the word “enzyme” in the abbreviation “ACE” Some background on the actual function of ACE2 in healthy cells should be presented (not just binding to SARS Cov-2), to add to the readers’ insight.

Response:  We thank the reviewer for the positive note in our review. We have added the number “2” after the word Angiotensin Converting Enzyme in line 70 as suggested by the reviewer. We have added the following “Upon entering the respiratory system SARS-CoV-2 binds to the receptor, Angiotensin Converting Enzyme 2 (ACE2) present in epithelial and endothelial cells of the upper respiratory tract. ACE2 receptors are also present in other non-immune and immune cells present in various organs like the heart, brain, skeletal muscles, kidney, intestine and endothelial cells. ACE2 is part of the Renin Angiotensin System (RAS), well known for circulatory homeostasis where it plays a major role in balancing the levels of Angiotensin II (AngII) and Angiotensin-(1–7). ACE2 is involved locally in multiple biological processes which include inflammation, angiogenesis, cell proliferation, memory, sodium and water reabsorption, thrombosis and plaque rupture. Human ACE2 protein is a zinc metallopeptidase, which is a type I transmembrane glycoprotein and has a single extracellular catalytic domain that predominantly localizes at the plasma membrane. Primarily a membrane protein in its full length, it also exists in truncated soluble form with the role of the latter in COVID-19 unclear. The SARS-CoV2 virus binds with membrane ACE2 receptors with strong affinity and the infected cells undergo conformational changes as soon as the virus fuses into cell membranes [10,11].“ from lines 69-83.

Comment:  Line 79- The statement “….transmembrane serine protease cleaves it to S2…” is not clear. Previous statement says that S protein contains two subunits S1 and S2. Hence the two statements do not add up. Please rephrase for clarity.

Response:  We apologize for this mistake. As suggested by the reviewer, we have rephrased the sentence for better clarity as suggested “As soon as the S1 subunit binds with the ACE2 receptor of the host cell, a transmembrane serine protease TMPRSS2 cleaves it from S2, which undergoes conformational change, and fuses with the host cell [17-19]. Alternately, in cells expressing low TMPRSS2, once engulfed into an endosome, SARS-CoV-2 escapes from the endosome by activation of their spike protein by cathepsins (non-specific low pH activated proteases) and further fusion of the host cell with viral membrane. In both mechanism, the SARS-CoV2 releases its RNA into the host cell cytoplasm, which translates to form two polyproteins that help in the formation of a replication translation complex in a double membrane vesicle.” for clarity in lines 96-102.

Comment:  Line 80 – “SARS-Cov2 escapes from the endososme…”, but the previous statement says S2 is engulfed by endosomes. Rephrasing is required so that the message is clear to general audience.

Response:  We apologize for this mistake. As suggested by the reviewer, we have rephrased the sentence for better clarity as suggested “As soon as the S1 subunit binds with the ACE2 receptor of the host cell, a transmembrane serine protease TMPRSS2 cleaves it from S2, which undergoes conformational change, and fuses with the host cell [17-19]. Alternately, in cells expressing low TMPRSS2, once engulfed into an endosome, SARS-CoV-2 escapes from the endosome by activation of their spike protein by cathepsins (non-specific low pH activated proteases) and further fusion of the host cell with viral membrane. In both mechanism, the SARS-CoV2 releases its RNA into the host cell cytoplasm, which translates to form two polyproteins that help in the formation of a replication translation complex in a double membrane vesicle.” for clarity in lines 96-102.

Comment:  Line 88 – “….can trigger inflammation while inhibiting host antimicrobial responses.” Please explain what are the host microbial responses that are inhibited in SARS cov2

Response: We apologize for the lack of clarity in this sentence. As suggested by the reviewer, we have revised this sentence and added the following “Several SARS-CoV-2 proteins, including open reading frame 3b (ORF3b), ORF6, ORF7, ORF8, and the N protein, can trigger inflammation while inhibiting host antimicrobial responses, such as delaying the onset of type I (IFNα, IFNβ etc.) and type III (IFN λ, IFNε etc.) interferon responses against viral infection [23]” to lines 105-110.

Comment:  Lin 103 – “….as evidenced by the kinetics of various IgM, IgG and IgA antibodies”, please elaborate more on what it is meant by the kinetics of antibodies.

Response: We apologize for the lack of clarity in this sentence. As suggested by the reviewer, we have revised this sentence as follows “SARS-CoV-2 elicits a strong B-cell response as evidenced by the robust and rapid expression of various antibodies, including IgM, IgG and IgA, soon after infection that can be detected in the blood sample of infected individuals” in lines 125-127.

Comment:  Line 113 – “Studies from previous and current COVID-19 pandemics…” what were the differences in the two cases- please elaborate.

Response: We thank the reviewer for pointing this out. We have not compared the previous pandemic but only studies across last 5 years and hence rephrased the sentence as “Studies from current COVID-19 pandemics have thrown light on the host and viral responses in these RNA viruses [1,3,8,14,21]” and retained appropriate citations in lines 139-141.

Comment:  Line 122 – “…contained SNPs in the receptor binding domains (RBDs) of S proteins”, considering the RBD is a small domain, the SNPs in each variant compared to wildtype should be presented in a tabular form. Also include both VOI and VOC to show how they differ in the mutations.

Response: The SNPs are elaborated in a previous study and hence we have provided the same as “The SNPs in these VOCs and VOIs have been described previously [31,32]” in lines 153-154.

Comment:  Line 135 – “ …. Ritonavir increases the concentration of Nirmatrelvir.” Please explain what this “increase” means

Response: We have now added the information and rephrased as “While Nirmatrelvir is a protease inhibitor, Ritonavir, a strong cytochrome P450 (CYP) 3A4 inhibitor and pharmacokinetic boosting agent increases the concentration of Nirmatrelvir to target therapeutic range” in lines 163-165.

Comment:  The phrase “humoral and cell-mediated immune responses” have been used through out the manuscript. Early on please define what these responses include.

Response: We apologize for the lack of clarity in this sentence. As suggested by the reviewer, we have revised this sentence as follows: “Adaptive immune responses, mediated by both T cell (not-associated with antibodies but T- cell driven cytokines/chemokines) and B cells (associated with antibodies, such as IgG, IgA, IgM etc) resulting in cell-mediated immunity and humoral immunity, respectively, were observed in COVID-19 [16]” in lines 110-113.

Comment:  As the paper talks about the virus and its mode of infection, a figure summarizing this process is recommend.

Response: We thank the reviewer for this recommendation; however, since there are numerous figures summarizing the SARS-CoV-2 infection are already available in the literature and quickly accessed, we cited those references and described it in our review.

Round 2

Reviewer 2 Report

The authors made the suggested corrections, however it is necessary:

Check lines 80, 84, 100, SARS-CoV2.

Check line 221, (antigens)in

Check line 280, SARS-CoV-2[52-55]

Review lines 310 and 325, SARS-CoV-2 S protein

Check spelling and grammar.

Furthermore, it is necessary to improve this very general conclusion.

Minor editing of English language required

Author Response

POINT-BY-POINT RESPONSE TO REVIEWER COMMENTS

Comment: The authors made the suggested corrections, 

Response: We thank the reviewer for the acknowledgement

Comment: Check lines 80, 84, 100, SARS-CoV2.

Response: We thank the reviewer for pointing out this error. We corrected these lines as well as checked throughout the article to fix this issue in the revised version.

Comment: Check line 221, (antigens)in

Response: We thank the reviewer for pointing out this error. We corrected these lines as well as checked throughout the article to fix this issue in the revised version.

Comment: Check line 280, SARS-CoV-2[52-55]

Response: We thank the reviewer for pointing out this error. We corrected these lines as well as checked throughout the article to fix this issue in the revised version.

Comment: Review lines 310 and 325, SARS-CoV-2 S protein

Response: We thank the reviewer for pointing out this error. We corrected these lines as well as checked throughout the article to fix this issue in the revised version.

Comment: Check spelling and grammar.

Response: We checked throughout the article to fix any spelling and/or grammatical error.

Comment: Furthermore, it is necessary to improve this very general conclusion.

Response: As suggested by the reviewer, we have significantly revised the conclusion section in the revised version.